# Childhood food insecurity and incident asthma: A population-based cohort study of children in Ontario, Canada

Kristin K. Clemens [1,2,3,4,5]*, Britney Le[3], Alexandra M. Ouédraogo[3], Constance Mackenzie[5,6], Marlee Vinegar[7], Salimah Z. Shariff[3,8]

1 Division of Endocrinology and Metabolism, Department of Medicine, Western University, London, Ontario, Canada, 2 Department of Epidemiology and Biostatistics, Western University, London, Ontario, Canada, 3 ICES, Ontario, Canada, 4 Lawson Health Research Institute, London, Ontario, Canada, 5 St. Joseph's Health Care, London, Ontario, Canada, 6 Divisions of Respirology and Clinical Pharmacology and Toxicology, Department of Medicine, Western University, London, Ontario, Canada, 7 Schulich School of Medicine and Dentistry, Western University, London, Ontario, Canada, 8 Arthur Labatt Family School of Nursing, Western University, London, Ontario, Canada

* kristin.clemens@sjhc.london.on.ca

**Data Availability Statement:** The dataset from this study is held securely in coded form at ICES. While data sharing agreements prohibit ICES from making the dataset publicly available, access may

## Abstract

### Background

Childhood food insecurity has been associated with prevalent asthma in cross-sectional studies. Little is known about the relationship between food insecurity and incident asthma.

### Methods

We used administrative databases linked with the Canadian Community Health Survey, to conduct a retrospective cohort study of children <18 years in Ontario, Canada. Children without a previous diagnosis of asthma who had a household response to the Household Food Security Survey Module (HFSSM) were followed until March 31, 2018 for new asthma diagnoses using a validated administrative coding algorithm. We used multivariable Cox proportional hazard models to examine the association between food insecurity and incident asthma, and adjusted models sequentially for clinical and clinical/socioeconomic risk factors. As additional analyses, we examined associations by HFSSM respondent type, severity of food insecurity, and age of asthma diagnosis. Moreover, we assessed for interaction between food security and child's sex, household smoking status, and maternal asthma on the risk of incident asthma.

### Results

Among the 27,746 included children, 5.1% lived in food insecure households. Over a median of 8.34 years, the incidence of asthma was 7.33/1000 person-years (PY) among food insecure children and 5.91/1000 PY among food secure children (unadjusted hazard ratio [HR] 1.24, 95% CI 1.00 to 1.54, p = 0.051). In adjusted analyses associations were similar (HR 1.16, 95% CI 0.91 to 1.47, p = 0.24 adjusted for clinical risk factors, HR 1.24, 95% CI 0.97 to 1.60, p = 0.09 adjusted for clinical/socioeconomic factors). Associations did not

be granted to those who meet pre-specified criteria for confidential access, available at https://www.ices.on.ca/DAS. The full dataset creation plan and underlying analytic code are available in Supporting Information 2 and 3.

**Funding:** This study was completed at the ICES Western site. ICES is funded by an annual grant from the Ontario Ministry of Health and Long-Term Care (MOHLTC). Core funding for ICES Western is provided by the Academic Medical Organization of Southwestern Ontario (AMOSO), the Schulich School of Medicine and Dentistry (SSMD), Western University, and the Lawson Health Research Institute (LHRI). Parts of this material are based on data and information compiled and provided by the MOHLTC and CIHI. The analyses, conclusions, opinions and statements expressed herein are solely those of the authors and do not reflect those of the funding or data sources; no endorsement by ICES, AMOSO, SSMD, LHRI, Western University or the MOHLTC is intended or should be inferred.

**Competing interests:** Outside of this work, Kristin Clemens received a research award, sponsored in part by Astra Zeneca. She has attended Merck sponsored conferences. She has received honoraria for delivering Certified Medical Education talks from Sutherland Global Services Canada ULC, the Toronto Ontario Knowledge Translation Working Group Inc, and the Canadian Medical and Surgical Knowledge Translation Research Group. These disclosures do not alter our adherence to PLOS ONE policies on sharing data and materials. There are no other conflicts of interest to disclose.

qualitatively change by HFSSM respondent type, severity of food insecurity, and age of asthma diagnosis. There was no evidence of interaction in our models.

## Conclusions

Food insecure children have numerous medical and social challenges. However, in this large population-based study, we did not observe that childhood food insecurity was associated with an increased risk of incident asthma when adjusted for important clinical and socioeconomic confounders.

## Introduction

One in 6 Canadian children face hunger and dietary compromise from food insecurity (inadequate or insecure access to healthy food due to financial constraints) [1]. Where nutritious and sufficient food is crucial to the growth and development of children [2], it is essential to understand the long-term health impact of food insecurity on our youth.

Asthma is a common chronic health condition in both children and adolescents [3]. As a multifactorial disease, asthma is thought to have genetic and environmental influences [4]. Asthma has been linked with exposure to smoking, allergens [5], socioeconomic status, obesity [6], poor nutrition [7], and psychological stress [8]. An association between childhood food insecurity and prevalent asthma has also been proposed in previous observational studies. Prior studies however, have had several methodological limitations; they have been small in size, single-centered, cross-sectional in design, used unvalidated measures of food insecurity, and captured self- or parental reported health outcomes [9–11]. Moreover, studies did not fully consider important confounders that can impact food insecurity-asthma relationships including maternal health and smoking. There has additionally been limited investigation on whether childhood food insecurity is associated with *new* asthma diagnoses (i.e. incident disease).

In this study, we used linked health services databases, and household health survey data (Canadian Community Health Survey), to examine the association between childhood food insecurity and incident asthma in a large cohort of Canadian children. We hypothesized that there would be an independent association between food insecurity and incident asthma.

## Materials and methods

### Design and setting

We conducted a population-based cohort study of children <18 years living in Ontario, Canada. Ontario is Canada's most populous province (>14 million residents). Residents have universal access to health services including hospital and physician care. Use of health services is captured by administrative codes held in secure databases available for access at ICES (formerly the Institute for Clinical Evaluative Sciences). ICES is an independent, non-profit research institute whose legal status under Ontario's health information privacy law allows it to collect and analyze healthcare and demographic data without consent, for health system evaluation and improvement. Use of data in this project was authorized under section 45 of Ontario's Personal Health Information Protection Act, which does not require review by a Research Ethics Board. We followed the guidelines for the REporting of studies Conducted using Observational Routinely-collected Data (RECORD) (S1 Table) [12].

## Data sources

We used linked ICES administrative databases and the Canadian Community Health Survey as our data sources. Datasets were linked using unique encoded identifiers and analyzed at ICES. Our project protocol (i.e. dataset creation plan) and analytic code are included in S1 and S2 Files.

A description of ICES databases is provided in S2 Table. We captured vital statistics and demographics from the Registered Persons Database of Ontario. We assessed immigration status using the Immigration, Refugees and Citizenship Canada's Permanent Resident Database. We used the Ontario Marginalization Index (ON-MARG) database to present measures of marginalization. These measures included residential instability (e.g. living alone, living in multi-unit housing), material deprivation (e.g. low income, unemployment), dependency (e.g. age ≥65 years), and ethnic concentration (e.g. recent immigrant, visible minority) [13,14].

We also had access to a unique database (MOMBABY) which links mothers to biological children born in Ontario. MOMBABY allowed for capture of perinatal and maternal risk factors associated with asthma (e.g. maternal asthma). We abstracted additional child and maternal comorbidities using the Canadian Institute for Health Information's Discharge Abstract Database (contains data acquired during hospital visits), and the National Ambulatory Care Reporting System Database (data on emergency department or ED visits). Finally, we used the Ontario Health Insurance Plan (OHIP) database to present additional comorbidities and health services utilization (e.g. visits to doctor's offices, use of ED services).

The Canadian Community Health Survey (CCHS), is a national, cross-sectional survey that contains health status information, health services use, and determinants of health at the individual- and household- level [15]. We captured food security status using the CCHS' Household Food Security Survey Module (HFSSM). The HFSSM was adapted from a long-standing food insecurity questionnaire used in the United States [16]. Details are provided in S3 Table. There are 18 questions related to household access to food over the previous 12 months; 10 of 18 questions target adult food experiences in the household (adult scale). Remaining questions focus on the experiences of children (child scale). Based upon the results of the HFSSM, children can be classified into several food security categories:

1. Food secure: No or one indication of difficulty with income-related food access

2. Moderately food insecure: Indication of some compromise in quality and/or quantity of food consumed (i.e. 2 to 4 affirmative responses)

3. Severely food insecure: Reduced food intake and disrupted eating patterns (i.e. ≥5 affirmative responses) [17].

Since the initial inclusion of the HFSSM in the CCHS in 2004, it has been used in 5 additional CCHS cycles that are available for use at ICES (2005, 2007–2008, 2009–2010, 2011–2012, 2013–2014). In addition to food security status, the CCHS also allowed for capture of important socioeconomic risk factors for disease including household smoking status, number of children in the home, and highest level of household education.

## Cohort

Our child cohort was built using previously described methods [18]. In brief, we included children in three ways. We included: 1) children *whose mothers* completed the HFSSM, 2) children age 12–17 years who *personally* completed the HFSSM, and 3) children who were *siblings* of a child HFSSM respondent. We confirmed that children, siblings and mothers were alive and living in the same household when the HFSSM was completed (matched upon residential

postal codes). After identifying eligible children, we excluded those with a previous diagnosis of asthma to focus on the risk of incident disease (diagnosis of asthma detailed below).

## Exposure

Our primary exposure was food insecurity as defined by the child scale of the HFSSM. Given the small number of severely food insecure children in our cohort (n = 1079), we a-priori grouped both moderately and severely food insecure children into one "food insecure" category. A similar approach has been used in previous research studies on food insecurity, including our recently published report [18,19]. We defined unexposed children as those who were food secure as per the HFSSM.

## Characteristics

We examined the age, racial belonging, perinatal history (caesarian section, prematurity and intrauterine growth restriction [IUGR]), comorbidities (e.g. respiratory syncytial virus), and baseline healthcare utilization (e.g. physician visits) of included children. We also captured important maternal characteristics including maternal age on child's birthday, immigration status, and maternal comorbidities. Moreover, we present household socioeconomic characteristics including measures of marginalization (as defined by the ON-MARG database above), location of residence (urban vs. rural), household smoking status, income, number of children in the home, highest household level of education.

## Primary outcome

We followed children in our cohort from the date of their first household HFSSM survey within the study period (i.e. our index date) until March 31, 2018. We captured new diagnoses of asthma using the Ontario Asthma Cohort (ASTHMA). The ASTHMA Cohort defines someone as having asthma if they have had at least one hospital admission with an asthma diagnosis, or two or more OHIP claims with an asthma diagnosis within two years. In those ≥18 years, the sensitivity of the algorithm is 80.6%, specificity 81.4%, positive predictive value 72.5%, negative predictive value 87.3% compared with medical chart review. In those <18 years, the algorithm's sensitivity is 89% with a specificity of 72% compared with medical chart review [20]. The ASTHMA cohort has been widely used to understand the epidemiology and healthcare burden of asthma in our province [3,21,22].

As additional post-hoc outcomes, we examined the association between childhood food insecurity and asthma by HFSSM respondent type (i.e. self-responder vs. other, age of responder). We also examined the association between food insecurity and incident asthma using a 3-category exposure definition of food insecurity (i.e. secure, moderate and severely food insecure), and conducted a related trend analysis. Recognizing that asthma diagnoses at ages <3 years can be unreliable [23], we carried out an additional analysis where we restricted to asthma diagnoses made at age 3 years or older. Finally we examined for interactions between food security status and child's sex, maternal asthma, and household smoking status on the risk of incident asthma.

## Statistical analysis

We present the baseline characteristics of included children descriptively (means and standard deviations, medians and interquartile ranges, numbers and percentages). We contrast the characteristics between food secure and insecure children using t-tests and Chi-squared analyses.

The details of our time-to-event analysis have been reported previously [18]. In brief, we generated non-parametric Kaplan-Meier (KM) curves to compare the unadjusted incidence of asthma by exposure group. We then used marginal Cox proportional hazard models to adjust sequentially for clinical and clinical/socioeconomic confounders. We chose confounders based upon their known association with both food insecurity and asthma [24]. Our clinical confounders included race (white, black, other), history of prematurity and IUGR, health care utilization (visits to general practitioners/pediatricians, hospital or ED visits), maternal age at child's birth, maternal immigration status, maternal asthma status, and household smoking status [25–29]. Socioeconomic confounders included marginalization (i.e. level of dependency, instability as defined above), location of residence, home ownership status, single parent household, income status, number of children in the home, highest level of household education [25,30–32]. Prior to including these covariates in our Cox models, we assessed for collinearity using Variance Inflation Factors and Pearson's Correlation Coefficients.

Moreover, we examined for interactions between food security status and three variables (child's sex, maternal asthma status, household smoking status) using log likelihood ratio tests. To preserve statistical power, we conducted post-hoc analyses using our model adjusted for clinical confounders only.

## Results

Our study cohort inclusions and exclusions are provided in S5 Table. In total, there were 34,042 children with a household response to the HFSSM during the study period. After excluding 6,296 children with previously diagnosed asthma, there were 27,746 children left in our cohort (26,331 from food secure households and 1,415 from food insecure households). At the time of the index HFSSM survey, the mean age of included children was 8.7 years.

In Tables 1 and 2 we illustrate the characteristics of included children by food security status. Compared with food secure children, food insecure children were more often female, and were more commonly from ethnic minority groups. Food insecure children lived in lower income and marginalized neighbourhoods, rented housing, were from single parent families, and were more often exposed to smoking in the home. We also found that food insecure children more often had a history of IUGR, and that at baseline, they used more ED services than food secure children.

The characteristics of the 18,270 unique mothers of the children in our cohort are presented in Table 3. Compared with the mothers of food secure children, mothers of food insecure children were younger at the time of their child's birth, more often recent immigrants, and more frequently had asthma, obesity and diabetes than mothers of food secure children.

### Relationship between food insecurity and incident asthma

Children in our cohort were followed for a mean of 8.34 years (maximum duration of follow-up 13.24 year) for incident asthma. We found 1398 new diagnoses over follow-up. Children received diagnoses at a mean age of 8.90 years.

The rate of new asthma diagnoses/1000 person-years (PY) was higher in food insecure vs. secure children, but the association was not statistically significant (7.33/1000 PY vs. 5.91/1000 PY; HR 1.24, 95% CI 1.00 to 1.54, p = 0.051) (Table 4). A similar non-significant relationship was observed when adjusted for clinical (HR 1.09, 95% CI 0.85 to 1.4, p = 0.496) and clinical/socioeconomic confounders (HR 1.24, 95% CI 0.97 to 1.60, p = 0.09) (Tables 4 and S6). Of note there were no collinearity issues between any of the model covariates (Pearson's r < |0.5| and variance inflation factor < 4).

**Table 1. Baseline characteristics of food insecure and secure children in Ontario, Canada.**

| | Food Secure | Food Insecure | P-value |
|---|---|---|---|
| | N = 26,331 | N = 1,415 | |
| **Demographics of children** | | | |
| Age at HFSSM survey date | | | |
| Mean (SD) | 8.66 ± 5.27 | 8.83 ± 5.02 | 0.244 |
| Median (IQR) | 9 (4–13) | 9 (5–13) | |
| 0–3 years (pre-school) | 5,992 (22.8%) | 282 (19.9%) | 0.051 |
| 4–5 years (kindergarten) | 2,646 (10.0%) | 144 (10.2%) | |
| 6–13 years (grade/middle school) | 11,414 (43.3%) | 657 (46.4%) | |
| 14–17 years (high school) | 6,279 (23.8%) | 332 (23.5%) | |
| Female, N(%) | 13,363 (50.8%) | 759 (53.6%) | 0.034 |
| Income quintile, N(%) [a] | | | |
| Quintile 1 (lowest) | 3,763 (14.3%) | 465 (32.9%) | < .001 |
| Quintile 2 | 4,739 (18.0%) | 322 (22.8%) | |
| Quintile 3 | 5,658 (21.5%) | 299 (21.1%) | |
| Quintile 4 | 6,147 (23.3%) | 191 (13.5%) | |
| Quintile 5 (highest) | 5,961 (22.6%) | 137 (9.7%) | |
| Measures of marginalization | | | |
| Dependency, N(%) | | | |
| Quintile 1 (least dependent) | 5,449 (20.7%) | 270 (19.1%) | 0.019 |
| Quintile 2 | 5,446 (20.7%) | 305 (21.6%) | |
| Quintile 3 | 5,250 (19.9%) | 280 (19.8%) | |
| Quintile 4 | 5,185 (19.7%) | 248 (17.5%) | |
| Quintile 5 (most dependent) | 4,827 (18.3%) | 305 (21.6%) | |
| Deprivation, N(%) | | | |
| Quintile 1 (least deprived) | 6,014 (22.8%) | 108 (7.6%) | < .001 |
| Quintile 2 | 6,223 (23.6%) | 224 (15.8%) | |
| Quintile 3 | 5,483 (20.8%) | 257 (18.2%) | |
| Quintile 4 | 4,532 (17.2%) | 293 (20.7%) | |
| Quintile 5 (most deprived) | 3,905 (14.8%) | 526 (37.2%) | |
| Ethnic Concentration, N(%) | | | |
| Quintile 1 (least concentrated) | 6,648 (25.2%) | 355 (25.1%) | < .001 |
| Quintile 2 | 6,331 (24.0%) | 295 (20.8%) | |
| Quintile 3 | 5,670 (21.5%) | 225 (15.9%) | |
| Quintile 4 | 4,256 (16.2%) | 235 (16.6%) | |
| Quintile 5 (most concentrated) | 3,252 (12.4%) | 298 (21.1%) | |
| Instability, N(%) | | | |
| Quintile 1 (less instability) | 6,309 (24.0%) | 217 (15.3%) | < .001 |
| Quintile 2 | 6,231 (23.7%) | 236 (16.7%) | |
| Quintile 3 | 5,527 (21.0%) | 291 (20.6%) | |
| Quintile 4 | 5,021 (19.1%) | 356 (25.2%) | |
| Quintile 5 (most instability) | 3,069 (11.7%) | 308 (21.8%) | |
| Rural location, N (%) | 5,725 (21.7%) | 290 (20.5%) | 0.484 |
| Ethnic Origin | | | |
| European | 21,291 (80.9%) | 996 (70.4%) | < .001 |
| Chinese | 500 (1.9%) | 22 (1.6%) | 0.353 |
| South Asian | 830 (3.2%) | 62 (4.4%) | 0.011 |
| Other | 6,706 (25.5%) | 566 (40.0%) | < .001 |

(*Continued*)

**Table 1.** (Continued)

| | Food Secure | Food Insecure | P-value |
|---|---|---|---|
| | N = 26,331 | N = 1,415 | |
| Racial belonging | | | |
| White | 20,392 (77.4%) | 836 (59.1%) | < .001 |
| Black | 597 (2.3%) | 124 (8.8%) | < .001 |
| East/Southeast Asian | 901 (3.4%) | 46 (3.3%) | 0.73 |
| West Asian/Arab | 276 (1.0%) | 36 (2.5%) | < .001 |
| South Asian | 798 (3.0%) | 61 (4.3%) | 0.007 |
| Latin American | 201 (0.8%) | 17 (1.2%) | 0.069 |
| Other | 3,486 (13.2%) | 322 (22.8%) | < .001 |
| **Comorbidities** | | | |
| Obesity | ≤10 | ≤5 | 0.269 |
| Prematurity | 1,623 (6.2%) | 100 (7.1%) | 0.17 |
| Intrauterine growth restriction | 389 (1.5%) | 33 (2.3%) | 0.01 |
| RSV | 111 (0.4%) | 6 (0.4%) | 0.989 |
| C-Section delivery | 5,719 (21.7%) | 316 (22.3%) | 0.586 |
| **Health Services Utilization** | | | |
| Mean (SD) hospital encounters | 0.08 ± 0.33 | 0.07 ± 0.29 | 0.172 |
| Mean (SD) ED encounters | 0.44 ± 0.95 | 0.56 ± 1.15 | < .001 |
| Mean (SD) GP visits | 2.10 ± 2.77 | 1.97 ± 2.50 | 0.1 |
| Mean (SD) Pediatrician visits | 0.53 ± 1.67 | 0.59 ± 1.63 | 0.178 |
| Mean (SD) Respirologist visits | 0.00 ± 0.03 | 0.00 ± 0.06 | 0.042 |

Missing marginalization data was recorded as "3".

Cell sizes <6 are not presented in accordance with ICES privacy regulations.

Abbreviations: GP, general practitioner; ED, emergency department; RSV, respiratory syncytial virus.

[a] Neighborhood income per person equivalent is a household size-adjusted measure of household income, based upon 2006 census summary data at the dissemination area level, using person-equivalents implied by low income cut-offs. Quintiles are defined within each area to reflect the relative nature of this measure, and to ensure that each area has about an equal percentage of the population in each income quintile [33].

When we examined the relationship between food insecurity and incident asthma by respondent type, there was also no statistically significant relationship observed (S7 Table). After excluding asthma diagnoses made in children <3 years, we noted similar non-significant results (HR 1.09, 95% CI 0.85 to 1.40, p = 0.479) (S8 Table). Although our analysis was limited by a small number of severely food insecure children, there did appear to be a statistically significant association between severe food insecurity and incident asthma (HR 2.32, 95% CI 1.1 to 4.9, p = 0.028) (S9 Table). However, when we conducted a related trend analysis, there was no significant association observed (HR 1.18, 95% CI 0.94 to 1.49, p = 0.132) (S10 Table). Finally, we did not find evidence of interaction between food security status and child's sex, maternal asthma status, nor household smoking on the risk of incident asthma (p values 0.208, 0.329, 0.306 respectively).

## Discussion

Food insecure children face social and health challenges including poor nutrition, poverty and obesity [35–37]. Previous reports (mainly cross-sectional) also suggest that food insecurity and prevalent asthma are related. For example, in a Canadian study of 9,142 children (10–15 years) and youth (16–21 years) who had measures of hunger ascertained (Canadian National

**Table 2. Household characteristics of food insecure and secure children in Ontario, Canada.**

| | Food Secure | Food Insecure | P-value |
|---|---|---|---|
| | N = 26,331 | N = 1,415 | |
| Household Food Security Status | | | |
| Food Secure | 23,359 (88.7%) | 0 (0.0%) | < .001 |
| Moderate Food Insecurity | 2,764 (10.5%) | 826 (58.4%) | |
| Severe Food Insecurity | 200 (0.8%) | 563 (39.8%) | |
| Smoking in home N (%) | | | |
| Yes | 2,217 (8.4%) | 335 (23.7%) | < .001 |
| Unknown | 7 (0.0%) | 0 (0.0%) | |
| Home ownership N (%) | | | |
| Yes | 22,630 (85.9%) | 674 (47.6%) | < .001 |
| Unknown | < = 70 | < = 5 | |
| Single parent household N (%) | | | |
| Yes | 3,485 (13.2%) | 604 (42.7%) | < .001 |
| Unknown | 2,025 (7.7%) | 125 (8.8%) | |
| Distribution of household income in deciles N (%) [a] | | | |
| 1 (lowest) | 1,911 (7.3%) | 618 (43.7%) | < .001 |
| 2 | 2,202 (8.4%) | 260 (18.4%) | |
| 3 | 2,354 (8.9%) | 213 (15.1%) | |
| 4 | 2,662 (10.1%) | 119 (8.4%) | |
| 5 | 2,754 (10.5%) | 70 (4.9%) | |
| 6 | 2,974 (11.3%) | 49 (3.5%) | |
| 7 | 2,939 (11.2%) | 12 (0.8%) | |
| 8 | 2,650 (10.1%) | 13 (0.9%) | |
| 9 | 2,701 (10.3%) | 8 (0.6%) | |
| 10 (highest) | < = 1,800 | < = 5 | |
| Unknown | < = 1,390 | < = 50 | |
| Number of children in household N (%) | | | |
| 1 | 5,829 (22.1%) | 279 (19.7%) | < .001 |
| 2 | 12,711 (48.3%) | 558 (39.4%) | |
| 3 | 5,627 (21.4%) | 361 (25.5%) | |
| 4+ | 2,164 (8.2%) | 217 (15.3%) | |
| Highest level of household education | | | |
| Less than secondary | 735 (2.8%) | 128 (9.0%) | < .001 |
| Post-Secondary | 3,244 (12.3%) | 412 (29.1%) | |
| Certificate | 11,777 (44.7%) | 644 (45.5%) | |
| Bachelor's Degree or higher | 9,077 (34.5%) | 132 (9.3%) | |
| Unknown | 1,498 (5.7%) | 99 (7.0%) | |

Cell sizes <6 are not presented in accordance with ICES privacy regulations.

[a] Neighborhood income per person equivalent is a household size-adjusted measure of household income, based upon 2006 census summary data at the dissemination area level, using person-equivalents implied by low income cut-offs. Quintiles are defined within each area to reflect the relative nature of this measure, and to ensure that each area has about an equal percentage of the population in each income quintile [33].

Longitudinal Survey of Children and Youth survey), youth with repeated episodes of hunger had a higher odds of self-reported chronic conditions including asthma in adjusted analysis [9]. In a Brazilian study of 1,307 children aged 6–12 years from public elementary schools who had a response on the Brazilian food security scale, there was also a statistically significant

**Table 3.  Maternal characteristics of food insecure and secure children in Ontario, Canada.**

| | Food Secure | Food Insecure | P-value |
|---|---|---|---|
| | N = 17,349 | N = 921 | |
| Age at child's birth | | | |
| Mean (SD) age (years) | 29.47 ± 5.10 | 27.48 ± 5.86 | < .001 |
| Median (IQR) age (years) | 30 (26–33) | 27 (23–32) | |
| Immigrant status [a] | | | |
| Recent Immigrant | 743 (2.8%) | 69 (4.9%) | < .001 |
| Longer-term immigrant | 1,330 (5.1%) | 112 (7.9%) | |
| Long-term resident | 24,258 (92.1%) | 1,234 (87.2%) | |
| Obesity | 143 (0.5%) | 21 (1.5%) | < .001 |
| Diabetes | 784 (3.0%) | 73 (5.2%) | < .001 |
| Charlson score [b] | | | |
| 0 | 25,736 (97.7%) | 1,375 (97.2%) | 0.374 |
| 1 | 304 (1.2%) | 21 (1.5%) | |
| 2+ | 291 (1.1%) | 19 (1.3%) | |
| Asthma | 3,125 (11.9%) | 290 (20.5%) | < .001 |

[a] Recent immigrant is defined as a person who landed officially as permanent resident <10 years prior to the interview date. A longer-term immigrant is defined as a person who landed officially as permanent resident 10–19 years prior to interview date. A long-term resident is defined as a person who landed officially as permanent resident > = 20 years prior to interview date.

[b] Charlson score is a weighted measure ranging from 0–31, which captures the relative effects of 17 different health conditions and is based on ICD-10 diagnostic codes. Each disease is assigned a value, and the sum of the values produces an individual's Charlson score. The Charlson score provides a measure of expected mortality, rather than quality-of-life related morbidity. [34].

association between asthma (defined by self-reported wheezing in the prior 12 months) and moderate and severe food insecurity (moderate food insecurity OR 1.71, 95% CI 1.01 to 2.89; severe food insecurity OR 2.51, 95% CI 1.28 to 4.93) [11]. A United States study of 11,099 3rd grade children who completed the United States Department of Agriculture (USDA) 18-item HFSSM, noted that children in food insecure households had a 4% higher adjusted odds of asthma (95% CI 1.02 to 1.06), and that the odds of asthma doubled (OR 2.00, 95% CI 1.97 to

**Table 4.  Association between food security status and new diagnoses of asthma.**

| | Secure | Insecure | P-value |
|---|---|---|---|
| Number of Children | 26,331 | 1415 | - |
| Median (IQR) follow-up (years) | 8.34 (6.18–10.64) | 8.34 (5.9–10.88) | - |
| New asthma diagnoses [N (%)] | 1311 (4.98) | 87 (6.15) | - |
| Rate per 1000 person-year | 5.91 | 7.33 | - |
| Unadjusted HR (95% CI) | Ref | 1.24 (1.00 to 1.54) | 0.051 |
| Adjusted HR (95% CI) (clinical)[a] | Ref | 1.16 (0.91 to 1.47) | 0.235 |
| Adjusted HR (95% CI) (clinical/socioeconomic)[b] | Ref | 1.24 (0.97 to 1.60) | 0.089 |

[a] Clinical confounders included race, history of prematurity, intrauterine growth restriction, visits to general practitioners/pediatricians, hospital or ED visits, maternal age at child's birth, maternal immigration status, maternal asthma status, household smoking status.

[b] Clinical/socioeconomic confounders included the clinical confounders above, along with marginalization, location of residence, home ownership status, single parent household, household income, the number of children in the home, and their highest level of household education.

Abbreviations: IQR, interquartile range; HR, hazard ratio; PY, person-year.

2.03) in households that were both food-insecure and poor.(10) In a study of 6,731 children aged 13–14 who also completed the USDA 18-item HFSSM, household food insecurity in the year before kindergarten and in second grade was associated with a higher odds of parental-reported asthma (18% and 55% respectively) [38]. Food insecurity in the 2nd grade was also linked with a higher odds of asthma in the 5th and 8th grades (OR 1.55, 1.53 to 1.58).

There were several methodological limitations of these previous studies. In some studies >50% of children were lost to follow-up over time. Many did not adjust for maternal health or asthma status, perinatal risk factors, nor race, housing quality, smoking, income and education which have been linked with food insecurity [39]. It is known for example, that low income and low levels of education are linked with asthma in children [30] and that both factors are strongly associated with food insecurity.[28] There are racial disparities in asthma (black children are twice as likely as white children to have asthma),[40] and food security status. Poor housing quality has been linked with asthma, [25] and is more common in food insecure children. Maternal smoking and tobacco exposure in early life are amongst the strongest risk factors for childhood asthma, [31,32] and are also more common in food insecure children. Studies have also been cross-sectional and have not been able to illicit the association between food insecurity and incident asthma diagnoses.

In a large cohort of Ontarian children, we carefully adjusted for childhood, perinatal, maternal, and household confounders and did not observe a relationship between food insecurity and incident asthma diagnoses. To our knowledge, ours is the first large cohort study to investigate this association. Our study suggests that while food insecurity is an important determinant of health, it may not have an *independent* role in the pathogenesis of childhood asthma.

There are many strengths to our cohort study. We included a large number of children with universal access to healthcare and followed them for a median of 9 years. We used novel methods and unique data sources to increase the number of children in our cohort (linked children to their mothers and siblings). We focused on the risk of incident asthma, an outcome particularly important for chronic disease prevention efforts. Further, we used a validated coding algorithm for physician-diagnosed asthma, which is extremely well validated in both children and adults in our province. Moreover, we conducted this study in Canada, where children have full access to health services. This is extremely important as food insecure populations across other regions, might not have equal access to healthcare services. Additionally, although some analyses were limited by statistical power, we examined food insecurity-incident asthma relationships by HFSSM respondent type, severity of food insecurity, and age of asthma diagnoses.

Limitations of our study are that the CCHS does not include First Nations individuals, or members of the Canadian military [15]. We recognize that the CCHS is also subject to survey sampling bias, though we attempted to reduce bias by not only including children who completed the HFSSM, but those whose mothers and siblings completed the survey. Surveys are subject to response bias, and questions about food insecurity might be sensitive to answer. However, the HFSSM remains our gold standard measure of food security status in Canada [41].

Another limitation of our study is that we captured food security status at a single time point and it is possible that food security experiences may have changed over the course of follow-up. We used physician diagnosed asthma as our outcome, but recognize that we could not capture outcomes in children who did not use any health services over the study period. Although we examined for many confounders in our analyses, we could not include those that were not measurable with our data sources. These include environmental factors such as air pollution and home air quality, exposure to allergens, nutrient deficiency, breastfeeding during

infancy, and psychological stress and well-being [5,7,8,42–45]. Finally our study results are only fully generalizable to children living in Ontario, Canada.

In conclusion, although children who live with food insecurity have numerous clinical and socioeconomic challenges, in this large cohort study we did not find an independent association between food insecurity and incident asthma.

## Supporting information

**S1 Table. RECORD checklist of recommendations for the reporting of studies conducted using routinely collected health data.**
(DOCX)

**S2 Table. Description of Ontario health administrative databases.**
(DOCX)

**S3 Table. Description of the household food security survey module.**
(DOCX)

**S4 Table. Study covariates.**
(DOCX)

**S5 Table. Participant inclusions and exclusions.**
(DOCX)

**S6 Table. Association between food insecurity and incident asthma, adjusted for clinical and socioeconomic confounders.**
(DOCX)

**S7 Table. Association between childhood food insecurity and incident asthma by respondent type, adjusted for clinical confounders.**
(DOCX)

**S8 Table. Association between food insecurity and incident asthma, excluding asthma diagnoses at ages <3 years, adjusted for clinical confounders.**
(DOCX)

**S9 Table. Association between childhood food insecurity and asthma using 3-level exposure status, adjusted for clinical confounders.**
(DOCX)

**S10 Table. Association between childhood food insecurity and incident asthma using 3-category exposure treated as continuous variable, adjusted for clinical confounders.**
(DOCX)

**S1 File. Dataset creation plan.**
(DOCX)

**S2 File. Analytic code.**
(TXT)

## Acknowledgments

We thank the Immigration, Refugees and Citizens of Canada for access to their Permanent Resident's Database.

## Author Contributions

**Conceptualization:** Kristin K. Clemens, Britney Le, Constance Mackenzie, Salimah Z. Shariff.

**Data curation:** Britney Le, Alexandra M. Ouédraogo, Salimah Z. Shariff.

**Formal analysis:** Britney Le, Alexandra M. Ouédraogo, Salimah Z. Shariff.

**Investigation:** Kristin K. Clemens, Marlee Vinegar, Salimah Z. Shariff.

**Methodology:** Kristin K. Clemens, Britney Le, Alexandra M. Ouédraogo, Salimah Z. Shariff.

**Project administration:** Kristin K. Clemens.

**Supervision:** Kristin K. Clemens.

**Writing – original draft:** Kristin K. Clemens.

**Writing – review & editing:** Britney Le, Alexandra M. Ouédraogo, Constance Mackenzie, Marlee Vinegar, Salimah Z. Shariff.

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
