## [Decision Letter · Decision Letter 0]

7 Dec 2020

PONE-D-20-29752

Childhood food insecurity and incident asthma: a population-based cohort study of children in Ontario, Canada

PLOS ONE

Dear Dr. Clemens,

Thank you for submitting your manuscript to PLOS ONE. After careful consideration, we feel that it has merit but does not fully meet PLOS ONE’s publication criteria as it currently stands. Therefore, we invite you to submit a revised version of the manuscript that addresses the points raised during the review process.

We look forward to receiving your revised manuscript.

Kind regards,

Maria Christine Magnus, MPH

Academic Editor

PLOS ONE

Journal Requirements:

2. Thank you for submitting the above manuscript to PLOS ONE. During our internal evaluation of the manuscript, we found some minor occurrences of overlapping text with the following previous publication(s), some of which you are an author, which needs to be addressed:

- https://onlinelibrary.wiley.com/doi/abs/10.1111/dme.14396

Please revise the manuscript to quote or rephrase the duplicated text and cite your sources for text outside the methods section. Please note that further consideration is dependent on the submission of a manuscript that addresses these concerns about the overlap in text with published work

'Outside of this work, KC received a research award, sponsored in part by Astra Zeneca. She has attended Merck sponsored conferences. She has received honoraria for delivering Certified Medical Education Talks from Sutherland Global Services Canada ULC, the Toronto Ontario Knowledge Translation Working Group Inc and the Canadian Medical and Surgical Knowledge Translation Group. There are no other conflicts of interest to disclose.'

a. Please confirm that this does not alter your adherence to all PLOS ONE policies on sharing data and materials, by including the following statement: "This does not alter our adherence to  PLOS ONE policies on sharing data and materials.” (as detailed online in our guide for authors http://journals.plos.org/plosone/s/competing-interests).  If there are restrictions on sharing of data and/or materials, please state these.

Please note that we cannot proceed with consideration of your article until this information has been declared.

Reviewers' comments:

Reviewer's Responses to Questions

**Comments to the Author**

1. Is the manuscript technically sound, and do the data support the conclusions?

Reviewer #1: Partly

Reviewer #2: Partly

2. Has the statistical analysis been performed appropriately and rigorously? 

Reviewer #1: I Don't Know

Reviewer #2: Yes

3. Have the authors made all data underlying the findings in their manuscript fully available?

Reviewer #1: Yes

Reviewer #2: Yes

4. Is the manuscript presented in an intelligible fashion and written in standard English?

Reviewer #1: Yes

Reviewer #2: Yes

5. Review Comments to the Author

Reviewer #1: Study is interesting but I have some concerns related to covariates and findings.

1. It is not very clear which covariates were included in the adjusted models. How they were chosen for the final models? Not all covariates are real confounders and models might be “overfitted”. You should consider using some methods for choosing confounders, like causality graphs. For example, when comorbidities were defined and were they included in the final model? They might be consequences of food insecurity and should not be included as confounders. What are clinical confounders and socioeconomic confounders? They should be explained as footnotes in the main tables and also in the text. How covariates were grouped or removed (page 15)?

2. You report that the association between food insecurity and incidence of asthma is non-significant after adjustment for all the confounders. However, in S7 Table you report significant adjusted hazard ratio for severe food insecurity (vs. food secure) and a dose-response effect for 3-category definition of food insecurity. Why this finding is not reported as a main finding? P value for trend should be reported. It seems that the study conclusions are inconsistent with the results.

3. Please, combine T4 and T5 and explain clinical and socioeconomic confounders in the footnote.

4. In Table 1, study characteristics are presented against exposure variable and in S9 Table against outcome variable. They should be presented similar way in both tables.

5. You present results by respondent type but other possible modifying effects should also be investigated, such as effect of sex and maternal asthma. Also interactions should be investigated.

6. In the Discussion page 22, you write that your findings differ from previous studies. Please, rewrite this paragraph after considering the suggestions above.

Reviewer #2: The manuscript addresses an interesting topic, the relationship between food security and the incidence of childhood asthma. Several methodologic considerations and the overall lack of detail dampen my enthusiasm for the analysis as currently presented.

Major points:

I suggest presenting a biologic mechanism that justifies an induction period of 12 months, particularly since the majority of children had moderate food insecurity. Along those lines, I would have like to have the definition of secure, moderate and severe food insecurity.

Rationale for combining moderate and severe food insecurity in terms of asthma risk. The citations used to justify the exposure variable addressed mental health and health care costs, not asthma.

The age range includes children < 3 years old where a diagnosis of asthma is unreliable. This is a major limitation of the manuscript.

Not enough information is given on the data sources. For example – what are the coverage rates? Is it reasonable to think that families of low income status and recent immigrants would be less likely to have completed the CCHS. No information on coverage rates for any data source were provided.

Important risk factors are missing (environmental, psychosocial, housing quality, etc) and these limitations should be highlighted more. History of respiratory infection was mentioned as a covariate but it appears the variable was actually RSV specifically.

There wasn’t enough information in the manuscript describing the covariates. I find it awkward to have to go to a supplemental table to obtain variable definitions. I suggest defining the covariates in the manuscript and the data source. For example: social deprivation is very important but I can’t judge its validity due to lack of information. Many covariates that appear in the table are not mentioned in the text.

I believe there is a typo in the Results ‘26,336 food insecure’ - I think they mean secure. ‘1,414 secure’ = insecure

Table 1: for variables where quintiles are presented, at least present the data range so the reader has an idea of what the levels are.

Table 1: there’s no definition for many variables. For example – dependency, deprivation, instability, obesity.

Table 2. No need to include both yes and no. Same comment as above for income. Added category of moderate food insecure which wasn’t mentioned in the exposure section. Overall, there are too many categories for several of the variables.

Table 3. Present units: like age in years, etc. There is no definition for immigrant status, charlson score.

Table 5. Tables should stand alone. What are the models adjusted for?

The section of prevalent asthma lacks any detail – most of the information is in the supplemental table. This is difficult to evaluate. I suggest including the essential information or, if there is not room, to remove it.

The literature review lacks consistent detail on all studies cited: add age range and location of all studies. Comparing this study, with an age range of <18 to other studies is problematic and the authors should discuss these differences in depth. Editorial suggestion: include age ranges, not just the grade level as this may change between countries. Also there’s no mention of how the exposure variables are similar or different between studies making it hard to decide if these studies are at all comparable.

Given the weaknesses of the study, I believe the conclusions are overstated.

6. PLOS authors have the option to publish the peer review history of their article (what does this mean?). If published, this will include your full peer review and any attached files.

Reviewer #1: No

Reviewer #2: No

---

## [Author Response · Author response to Decision Letter 0]

12 Apr 2021

Editor Comments

Comment 1. Please ensure that your manuscript meets PLOS ONE's style requirements, including those for file naming. The PLOS ONE style templates can be found at

Response 1. Thank you. We have re-reviewed PLOS ONE’s style requirements and have updated our title page and manuscript as suggested.

Comment 2. Thank you for submitting the above manuscript to PLOS ONE. During our internal evaluation of the manuscript, we found some minor occurrences of overlapping text with the following previous publication(s), some of which you are an author, which needs to be addressed:

- https://onlinelibrary.wiley.com/doi/abs/10.1111/dme.14396

Please revise the manuscript to quote or rephrase the duplicated text and cite your sources for text outside the methods section. Please note that further consideration is dependent on the submission of a manuscript that addresses these concerns about the overlap in text with published work

Response 2. We apologize for this. Both the current paper and our recently published manuscript were written at the same time, and upon re-review we completely agree that there is overlap in some sections. 

We have re-reviewed our manuscript in detail and have rephrased our prose throughout. In the Methods section, we now cite our recently published Diabetic Medicine paper, and have shorted our description of our cohort build and analytic methods. Please note that some of the phrasing used throughout our paper, does have to be written verbatim as per ICES policies. 

Comment 3. We note that you have indicated that data from this study are available upon request. PLOS only allows data to be available upon request if there are legal or ethical restrictions on sharing data publicly. For information on unacceptable data access restrictions, please see http://journals.plos.org/plosone/s/data-availability#loc-unacceptable-data-access-restrictions.

Response 3. There are legal and ethical restrictions that preclude us from sharing our data publicly. 

In S1 Table, we report: “The dataset from this study is held securely in coded form at ICES. While data sharing agreements prohibit ICES from making the dataset publicly available, access can be granted to those who meet pre-specified criteria for confidential access, available at https://www.ices.on.ca/DAS.The full data set creation plan and underlying analytic code are available in Supporting Information 2 and 3.” 

Comment 4: In your revised cover letter, please address the following prompts:

Response 4: In our revised cover letter, we have detailed the ethical and legal restrictions upon sharing de-identified ICES data. 

Comment 5. Thank you for stating the following in the Competing Interests section:

'Outside of this work, KC received a research award, sponsored in part by Astra Zeneca. She has attended Merck sponsored conferences. She has received honoraria for delivering Certified Medical Education Talks from Sutherland Global Services Canada ULC, the Toronto Ontario Knowledge Translation Working Group Inc and the Canadian Medical and Surgical Knowledge Translation Group. There are no other conflicts of interest to disclose.'

Response 5: We have provided our competing interests statement in our cover letter and have reported that “this does not alter our adherence to PLOS ONE policies on sharing data and materials”.

Reviewer comments

Reviewer #1: Study is interesting but I have some concerns related to covariates and findings.

Comment 1. It is not very clear which covariates were included in the adjusted models. How they were chosen for the final models? Not all covariates are real confounders and models might be “overfitted”. You should consider using some methods for choosing confounders, like causality graphs. For example, when comorbidities were defined and were they included in the final model? They might be consequences of food insecurity and should not be included as confounders. What are clinical confounders and socioeconomic confounders? They should be explained as footnotes in the main tables and also in the text. How covariates were grouped or removed (page 15)?

Response 1. Thank you. We conducted a thorough literature review on food insecurity and asthma and chose our covariates based upon their association with both our exposure (food insecurity) and our outcome (i.e. confounders). The covariates chosen also align with the variables used in our previously published paper. Based upon our review, we do not believe that covariates are direct consequences of food security alone, without also being associated with our exposure. 

In the Methods section on page 10, we now explicitly present the covariates used in each model. In S4 Table, more details are provided on how each confounder was treated and what our referent groups were. Prior to including covariates in our models, we examined for collinearity using Pearson r test and the VIF test. There were no collinearity issues found and thus no variables were removed. We have reported this in our Methods section on page 10, and in our Results section on page 15.

In our Tables, Supplementary Tables and footnotes, we have described how we grouped covariates

In our new Table 4, we have also listed all of the clinical and socioeconomic confounders we adjusted for as footnotes.

Comment 2. You report that the association between food insecurity and incidence of asthma is non-significant after adjustment for all the confounders. However, in S7 Table you report significant adjusted hazard ratio for severe food insecurity (vs. food secure) and a dose-response effect for 3-category definition of food insecurity. Why this finding is not reported as a main finding? P value for trend should be reported. It seems that the study conclusions are inconsistent with the results.

Response 2. This was a post-hoc exploratory analysis, and was unfortunately limited by a small number of severely food insecure children and events. However, in the revised manuscript, we have reported this result more explicitly on page 15. We have now also conducted a related trend analysis and added this to our results on page 15 and S10 Table of the Supplemental Material. The result of our trend analysis was consistent with or main results – that there was not a statistically significant relationship between childhood food insecurity and incident asthma.

Comment 3. Please, combine T4 and T5 and explain clinical and socioeconomic confounders in the footnote.

Response 3. We have now combined Tables 4 and 5 (new Table 4) and have included a description of all clinical and socioeconomic confounders in the footnotes.

Comment 4. In Table 1, study characteristics are presented against exposure variable and in S9 Table against outcome variable. They should be presented similar way in both tables.

Response 4. Our prevalent asthma analysis has been removed from this manuscript (per Reviewer 2’s comment). As such, S9 Table has been deleted from the resubmission.

Comment 5. You present results by respondent type but other possible modifying effects should also be investigated, such as effect of sex and maternal asthma. Also interactions should be investigated.

Response 5. We have now assessed for interaction between food insecurity and maternal asthma, child’s sex and smoking in the home. We have described our methods on page 10. We have presented results on page 15. The log-likelihood ratio test suggested no evidence of interaction. As such there were no interaction terms included in our models. 

Comment 6. In the Discussion page 22, you write that your findings differ from previous studies. Please, rewrite this paragraph after considering the suggestions above.

Response 6. We have re-written our Discussion to focus upon included results and have discussed study implications, strengths and limitations.

Reviewer #2: The manuscript addresses an interesting topic, the relationship between food security and the incidence of childhood asthma. Several methodologic considerations and the overall lack of detail dampen my enthusiasm for the analysis as currently presented.

Comment 1. I suggest presenting a biologic mechanism that justifies an induction period of 12 months, particularly since the majority of children had moderate food insecurity. Along those lines, I would have like to have the definition of secure, moderate and severe food insecurity.

Response 1. In the revised draft, we have moved our definitions of food security status to the main manuscript (Methods page 7).

We used a 12 month look-back for food insecurity status as this is the look-back period captured by the HFSSM. Both the CCHS’ HFSSM and the USDA’s HFSSM assess food security status in this fashion. We recognize that capturing food security status cross-sectionally is a limitation, and that the food security status could have changed as we followed our cohort over time. We have described this limitation in our Discussion on page 18.

Comment 2. Rationale for combining moderate and severe food insecurity in terms of asthma risk. The citations used to justify the exposure variable addressed mental health and health care costs, not asthma.

Response 2. There were only 1079 severely food insecure children in our cohort. Because we anticipated a limited number of outcomes, we elected a-priori combine moderate and severely food insecure children into one category (food insecure). We provided the mental health/healthcare costs paper as a reference, as authors grouped exposure categories similarly. 

Of note, we did conduct additional analyses in the current paper, where we examined relationships using a 3-category definition of food insecurity (Table S9-S10) and the results of our analyses were similar to our main findings.

Comment 3. The age range includes children < 3 years old where a diagnosis of asthma is unreliable. This is a major limitation of the manuscript.

Response 3. Thank you for this important comment. As a post-hoc sensitivity analyses, we examined associations where asthma diagnoses at ages <3 years were excluded. We found results similar to our primary analysis. We report our methods on page 10, and our results on page 15 and in Table S8.

Comment 4. Not enough information is given on the data sources. For example – what are the coverage rates? Is it reasonable to think that families of low income status and recent immigrants would be less likely to have completed the CCHS. No information on coverage rates for any data source were provided.

Response 4. Given our provinces universal healthcare system, all residents with an Ontario healthcard who sought healthcare are included in ICES databases. We do recognize that those who did not receive healthcare will not be captured in this study and have commented upon this in our Discussion on page 19. 

We also recognize that we had to exclude children who did not have a household response to the HFSSM (numbers provided in S5 Table). We also comment in our discussion that there may be sampling bias in our study due to use of the CCHS survey. However, we attempted to reduce the risk of sampling by not only including children who completed the CCHS, but also children of mothers who completed the CCHS. These points have been discussed in detail on page 18-19 of the Discussion.

Comment 5. Important risk factors are missing (environmental, psychosocial, housing quality, etc.) and these limitations should be highlighted more. History of respiratory infection was mentioned as a covariate but it appears the variable was actually RSV specifically.

Response 5. Thank you. We recognize that asthma is a multifactorial disease with clinical, environmental, and socioeconomic risk factors. We captured as many as able (maternal comorbidities, perinatal history, child comorbidities, SES, smoking, household characteristics), but recognize that some were not “measurable” for us to adjust for. We have expanded upon your point in our Discussion on page 19. 

You are indeed correct, we captured RSV infection rather than respiratory infection and have corrected this in our Methods section on page 8. 

Comment 6. There wasn’t enough information in the manuscript describing the covariates. I find it awkward to have to go to a supplemental table to obtain variable definitions. I suggest defining the covariates in the manuscript and the data source. For example: social deprivation is very important but I can’t judge its validity due to lack of information. Many covariates that appear in the table are not mentioned in the text.

Response 6. We have now provided the details of all covariates early in the Methods section on page 8. Those included in our adjusted models are reported on page 9-10. We have also included covariates included in each model in the footnotes of Table 4. We still present S4 Table as it might help readers to understand how we treated each covariate and what our referent group was for each.

Comment 7. I believe there is a typo in the Results ‘26,336 food insecure’ - I think they mean secure. ‘1,414 secure’ = insecure

Response 7. We apologize for this typo and have corrected this.

Comment 8. Tables

Table 1: for variables where quintiles are presented, at least present the data range so the reader has an idea of what the levels are.

Table 1: there’s no definition for many variables. For example – dependency, deprivation, instability, obesity.

Table 2. No need to include both yes and no. Same comment as above for income. Added category of moderate food insecure which wasn’t mentioned in the exposure section. Overall, there are too many categories for several of the variables.

Table 3. Present units: like age in years, etc. There is no definition for immigrant status, charlson score.

Table 5. Tables should stand alone. What are the models adjusted for?

Response 8. We have updated all Tables based upon your helpful feedback. Unfortunately we are not able to provide a range for quintiles. This is because quintiles differ by neighborhood, and so a quintile of 5 can be very different in Toronto vs. Windsor. We have included an explanatory footnote in Tables 1 and 2. 

In the Methods Section we provide more details on the Ontario Marginalization Database including what measures of dependency, deprivation etc. mean. 

We have included yes rather than both yes and no.

We have removed the moderate food insecurity category in Table 2. 

We have presented units like age in years.

We have also provided definitions for immigrant status and Charlson score as footnotes in Table 3.

We have merged Tables 4 and 5 as suggested and in the footnotes have reported what the models were adjusted for.

Comment 9. The section of prevalent asthma lacks any detail – most of the information is in the supplemental table. This is difficult to evaluate. I suggest including the essential information or, if there is not room, to remove it.

Response 9. We have elected to remove our prevalent asthma analysis from this paper.

Comment 10. The literature review lacks consistent detail on all studies cited: add age range and location of all studies. Comparing this study, with an age range of <18 to other studies is problematic and the authors should discuss these differences in depth. Editorial suggestion: include age ranges, not just the grade level as this may change between countries. Also there’s no mention of how the exposure variables are similar or different between studies making it hard to decide if these studies are at all comparable.

Response 10. We have re-written this section for consistency and have added more detail to help to illustrate the comparability of studies.

---

## [Decision Letter · Decision Letter 1]

14 May 2021

Childhood food insecurity and incident asthma: a population-based cohort study of children in Ontario, Canada

PONE-D-20-29752R1

Dear Dr. Clemens,

We’re pleased to inform you that your manuscript has been judged scientifically suitable for publication and will be formally accepted for publication once it meets all outstanding technical requirements.

Kind regards,

Maria Christine Magnus, MPH

Academic Editor

PLOS ONE

Additional Editor Comments (optional):

Reviewers' comments:

Reviewer's Responses to Questions

**Comments to the Author**

1. If the authors have adequately addressed your comments raised in a previous round of review and you feel that this manuscript is now acceptable for publication, you may indicate that here to bypass the “Comments to the Author” section, enter your conflict of interest statement in the “Confidential to Editor” section, and submit your "Accept" recommendation.

Reviewer #1: All comments have been addressed

2. Is the manuscript technically sound, and do the data support the conclusions?

Reviewer #1: Yes

3. Has the statistical analysis been performed appropriately and rigorously? 

Reviewer #1: Yes

4. Have the authors made all data underlying the findings in their manuscript fully available?

Reviewer #1: No

5. Is the manuscript presented in an intelligible fashion and written in standard English?

Reviewer #1: Yes

6. Review Comments to the Author

Reviewer #1: Thank you for your response. I think you addressed all my comments and I have no further comments to add.

7. PLOS authors have the option to publish the peer review history of their article (what does this mean?). If published, this will include your full peer review and any attached files.

Reviewer #1: No

---

## [Editor Report · Acceptance letter]

20 May 2021

PONE-D-20-29752R1 

Childhood food insecurity and incident asthma: a population-based cohort study of children in Ontario, Canada 

Dear Dr. Clemens:

I'm pleased to inform you that your manuscript has been deemed suitable for publication in PLOS ONE. Congratulations! Your manuscript is now with our production department. 

Kind regards, 

on behalf of

Dr. Maria Christine Magnus 

Academic Editor

PLOS ONE